

# Segmentation of multi-temporal polarimetric SAR data based on mean-shift and spectral graph partitioning

Caiqiong Wang[1,2,3], Lei Zhao[2,3], Wangfei Zhang[1], Xiyun Mu[4] and Shitao Li[5]

[1] College of Forestry, Southwest Forestry University, Kunming, Yunnan, China
[2] Institute of Forest Resource Information Techniques, Chinese Academy of Forestry, Beijing, China
[3] Key Laboratory of Forestry Remote Sensing and Information System, NFGA, Beijing, China
[4] Institute of Chifeng Forestry Research, Chifeng, Inner Mongolia, China
[5] College of Geography and Ecotourism, Southwest Forestry University, Kunming, Yunnan, China

Corresponding author
Lei Zhao, zhaolei@ifrit.ac.cn

## ABSTRACT

**Abstract:** Polarimetric SAR (PolSAR) image segmentation is a key step in its interpretation. For the targets with time series changes, the single-temporal PolSAR image segmentation algorithm is difficult to provide correct segmentation results for its target recognition, time series analysis and other applications. For this, a new algorithm for multi-temporal PolSAR image segmentation is proposed in this paper. Firstly, the over-segmentation of single-temporal PolSAR images is carried out by the mean-shift algorithm, and the over-segmentation results of single-temporal PolSAR are combined to get the over-segmentation results of multi-temporal PolSAR images. Secondly, the edge detectors are constructed to extract the edge information of single-temporal PolSAR images and fuse them to get the edge fusion results of multi-temporal PolSAR images. Then, the similarity measurement matrix is constructed based on the over-segmentation results and edge fusion results of multi-temporal PolSAR images. Finally, the normalized cut criterion is used to complete the segmentation of multi-temporal PolSAR images. The performance of the proposed algorithm is verified based on three temporal PolSAR images of Radarsat-2, and compared with the segmentation algorithm of single-temporal PolSAR image. Experimental results revealed the following findings: (1) The proposed algorithm effectively realizes the segmentation of multi-temporal PolSAR images, and achieves ideal segmentation results. Moreover, the segmentation details are excellent, and the region consistency is good. The objects which can't be distinguished by the single-temporal PolSAR image segmentation algorithm can be segmented. (2) The segmentation accuracy of the proposed multi-temporal algorithm is up to 86.5%, which is significantly higher than that of the single-temporal PolSAR image segmentation algorithm. In general, the segmentation result of proposed algorithm is closer to the optimal segmentation. The optimal segmentation of farmland parcel objects to meet the needs of agricultural production is realized. This lays a good foundation for the further interpretation of multi-temporal PolSAR image.

## INTRODUCTION

Polarimetric SAR (PolSAR) is an advanced remote sensing earth observation system. It detects ground targets by emitting and receiving electromagnetic waves in different polarimetric states, and can obtain rich scattering information of targets (*Ersahin, Cumming & Ward, 2010*). PolSAR image segmentation is a key step in its interpretation, and its performance directly affects the subsequent processing such as feature extraction, target recognition, ground target classification and so on (*Yu, 2012*). Therefore, the research on PolSAR image segmentation algorithm has always been the focus and hotspot of PolSAR image processing technology.

In recent years, numerous studies have put forward many segmentation algorithms according to the characteristics of PolSAR images. According to the core technology used, the existing PolSAR segmentation methods are roughly divided into the following four categories: (1) Segmentation methods based on graph theory, such as spectral graph partitioning (SGP)/spectral clustering (*Ersahin, Cumming & Ward, 2010*), max-flow min-cut method (*Zhou et al., 2020*), etc.; (2) Segmentation method based on region growth, such as simple linear iterative clustering (*Zou et al., 2016*), simple nonlinear iterative clustering (*Ma et al., 2021*), watershed algorithm (*Marcin, 2017*), etc.; (3) Segmentation method based on threshold, such as simple threshold division method (*Otsu, 2007*), maximum between-class variance (*Shao et al., 2013*), etc.; (4) Segmentation method based on specific theory, such as markov random fields (*Duan et al., 2017*), level set (*Mohammadimanesh et al., 2019*), neural network (*Fowlkes et al., 2004*), etc. Among them, the methods based on SGP have become a research hotspot in recent years due to its advantages of clustering in arbitrary-shaped sample spaces and converging to the global optimal solution, but its computational time and space costs are relatively high. Many scholars have carried out research on this problem, and some of them use sampling approximation techniques, such as Nyström sampling method proposed by *Fowlkes et al. (2004)*, which can effectively improve the efficiency of SGP, but the implementation process of this algorithm is complicated and the effect is not ideal for images with low signal-to-noise ratio. Other scholars solve this problem through combined segmentation algorithm. For example, the segmentation algorithm that combines watershed and spectral clustering proposed by *Ma & Jiao (2008)*; PolSAR image segmentation algorithm based on mean-shift (MS) and SGP (MS-SGP) proposed by *Zhao et al. (2015)*. This kind of combination algorithm usually first uses a local optimization algorithm to quickly realize the over-segmentation of PolSAR image, and then the global optimization of SGP algorithm is used to achieve excellent segmentation effect. Since the SGP is based on the over-segmentation regions (super-pixel) instead of the original pixels, the computational cost of SGP is effectively reduced. This kind of combination algorithm combines the advantages and disadvantages of local and global optimization segmentation

algorithms, and is the research focus of PolSAR image segmentation algorithms in recent years.

However, most of the existing PolSAR image segmentation algorithms are only for single-temporal PolSAR images. For the targets with time series changes, single-temporal PolSAR image can't provide enough information for segmentation. For example, crops of different types and different sowing times are easy to show similar characteristics on single-temporal PolSAR images, so it is impossible to segment field parcel objects that meet the needs of agricultural production based on single-temporal PolSAR images. Based on multi-temporal PolSAR images, the scattering characteristics of crops in different growth periods can be obtained, which is expected to solve the above problems.

In recent years, some scholars have begun to explore multi-temporal PolSAR image segmentation algorithms. For example, *Zou (2015)* proposed a variational level set segmentation algorithm for multi-temporal PolSAR images, which can effectively improve the recognition accuracy of ground targets. *Deng et al. (2014)* obtained more robust and accurate segmentation results by combining stationary wavelet transform and algebraic multigrid method for hierarchical segmentation of multi-temporal PolSAR images. *Ma et al. (2021)* realized super-pixel cooperative segmentation of dual-temporal SAR images through simple nonlinear iterative clustering. *Alonso-Gonzalez, Lopez-Martinez & Salembier (2014)* proposed a data processing method of PolSAR time series based on binary partition trees. Overall, there are relatively few segmentation algorithms suitable for multi-temporal PolSAR images. Moreover, most segmentation algorithms are based on local optimization to process the image, which will produce super-pixel regions and can't achieve optimal segmentation. However, as far as agricultural application is concerned, as the basic farming unit of agricultural production, accurate identification of farmland parcels is conducive to the realization of crop production planning, management and benefit evaluation. In addition, the polarization features extracted from farmland parcel units can avoid the influence of outliers, and are more accurate than the polarization features extracted from over-segmented objects based on homogeneous pixel clustering. Thus, it is more beneficial to the application of crop target identification and time series analysis. Obviously, the super-pixels produced by over-cutting can't meet the above production needs. Therefore, an optimal segmentation algorithm that can segment the basic units of farmland parcels is needed. At this time, based on multi-temporal PolSAR images and combined with the advantage that SGP algorithm can cluster in the sample space of arbitrary shape and converge to the global optimal solution, it is expected to achieve optimal segmentation and solve the above problems.

In view of the above analysis, a new algorithm suitable for multi-temporal PolSAR image segmentation is proposed based on the MS-SGP segmentation algorithm of single-temporal PolSAR image (*Zhao et al., 2015*). On the basis of combining the advantages and disadvantages of local optimization and global optimization segmentation algorithm, this algorithm comprehensively uses the rich polarimetric and temporal information of multi-temporal PolSAR images to realize image segmentation. The algorithm combines the over-segmentation results of single-temporal PolSAR images generated by MS algorithm. Furthermore, the edge information extracted by edge detector is fused. In order to

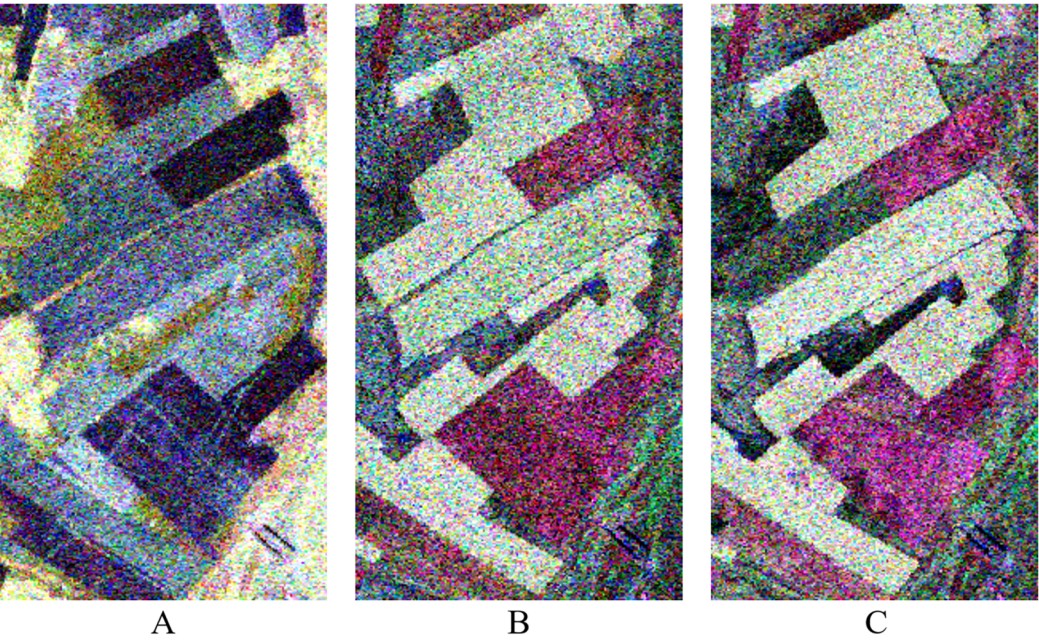

**Figure 1 Pauli RGB display of PolSAR data at different temporal in experimental region.** The data are three temporal PolSAR images of Radarsat-2 (C-band) acquired on May 23, August 3, and August 27, 2013, respectively. (A) 20130523, (B) 20130803, (C) 20130827.

obtain the over-segmentation result and edge fusion result of the multi-temporal PolSAR image. Based on this, the similarity measurement matrix is constructed to realize the segmentation of multi-temporal PolSAR images. In order to verify the performance of this algorithm, it is applied to the segmentation of time series PolSAR images of Radarsat-2, and the segmentation effect and accuracy of single-temporal and multi-temporal PolSAR image segmentation algorithms are compared and analyzed.

# MATERIALS AND METHODS

## Materials

### PolSAR data and the test site

The data used in this experiment are a time series Radarsat-2 PolSAR (C-band) images acquired on May 23, August 3, and August 27, 2013, respectively. The acquired three images have the same sensor parameters such as the ascending orbit, the imaging mode of FQ18, the incident angle of 37.5°, and the azimuth and range resolution with 4.96 and 4.73 m, respectively. The study area is covered by the acquired is Shangkuli Farm (119°07′~121°49′E, 50°01′~53°26′N) between city Erguna and Genhe, Inner Mongolia. Part of the farmland region is selected as for our experiment. The selected farmland region was planted mainly with rape (*Brassica napus* L), wheat (*Triticum aestivum* L) and other crops. Fig. 1 shows the PolSAR image (PauliRGB) for the experiments. Due to the difference in factors such as height, moisture content, and appearance of the same crop at different crop growth stages, different backscattering coefficients of the same farmland
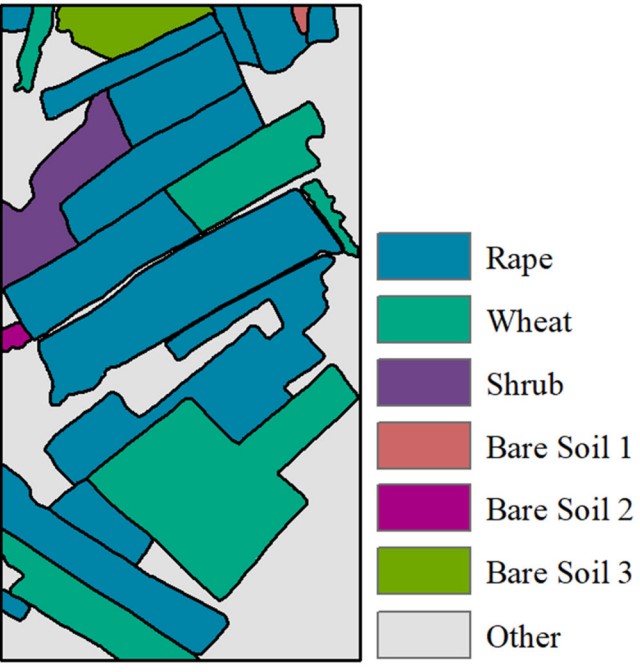

**Figure 2 Reference map for Segmentation Evaluation.**

parcel with same crop at different growth stages occurred in the time series images like in Fig. 1A, 1B & 1C. They show different color on multi-temporal PolSAR images.

### Reference map for segmentation evaluation

Reference map for segmentation evaluation, which is generated based on ground campaigned data and expert knowledge, is shown here as Fig. 2. In this study, the segmentation evaluation reference map will be used to evaluate the accuracy, so as to verify the performance of the proposed multi-temporal MS-SGP segmentation algorithm. For the generation process of Fig. 2, readers are referred to the details in *Zhao et al. (2015)*. There are mainly five types of land cover in this region, including rape, wheat, shrubs, bare soil and other types. Among them, bare soil shows three different scattering characteristics due to different factors such as roughness, ridge direction and water content, so it can be subdivided into three categories.

### Multi-temporal polarimetric SAR segmentation method

According to the data characteristics of multi-temporal PolSAR images, a multi-temporal MS-SGP segmentation algorithm suitable for multi-temporal PolSAR images is proposed on the basis of single-temporal MS-SGP segmentation algorithm. The technical flow is shown in Fig. 3. Firstly, the PolSAR data is pre-processed. Secondly, MS algorithm is used to pre-segment the image to obtain the initial segmentation unit, and the edge extraction algorithm is used to provide segmentation clues for SGP. Then, the similarity measurement matrix is constructed based on the segmentation unit and segmentation clues, and then the normalized cut criterion is used to complete the segmentation of image.

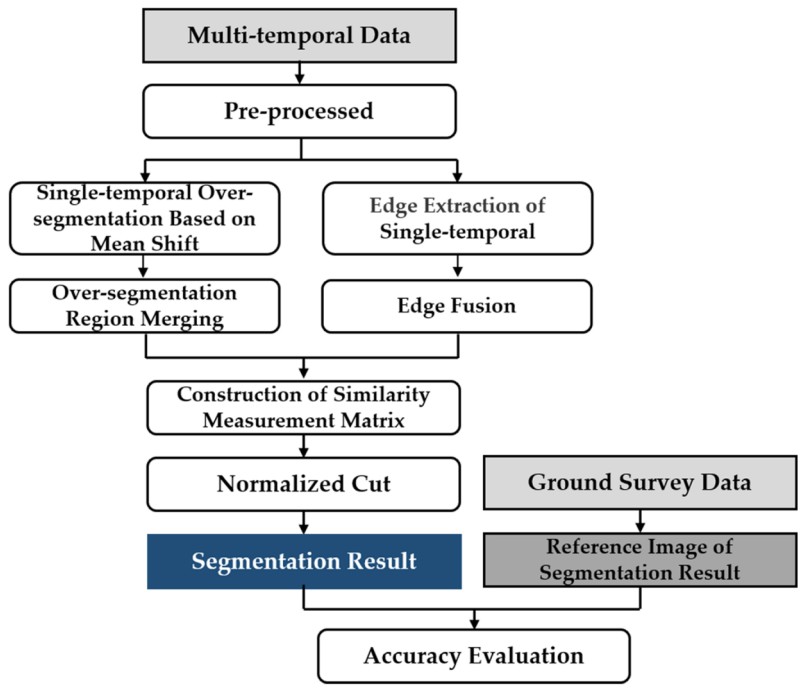

**Figure 3 Technical flow chart of multi-temporal MS-SGP segmentation algorithm.**

Finally, the segmentation results are evaluated in detail based on the reference segmentation map obtained from the ground survey data.

### Pre-segmentation

(1) MS method for single-temporal SAR image over-segmentation

MS is an iterative algorithm for nonparametric kernel density estimation (*Fukunaga & Hostetler, 1975*). Firstly, the offset mean value of the current point is calculated in the feature space, and the point is moved according to its offset mean value. Then, taking the location of the above moved point as the input of the new starting point and repeating the procedure until it converges to the convergence point of probability density function. The offset mean value is calculated as follows:

$$M_h(x) = \frac{\sum_{i=1}^{n} G[(x_i - x)/h] w(x_i - x)}{\sum_{i=1}^{n} G[(x_i - x)/h] w(x_i)} \tag{1}$$

where $h$ is the window size of the kernel function, $G$ is the kernel function, in this paper, Gaussian function is selected as kernel function. $w(x_i) \geq 0$ is weight function assigned to each sampling point $x_i$, $x = (x_r, x_s)$ where $x_s$ is a two-dimensional space vector, which include the coordinate information of each pixel, $x_r$ is the feature vector (*He et al., 2008*). Three polarimetric scattering components decomposed by PolSAR Pauli are used as the features vector of $x_r$. The kernel function is shown as Eq. (2).

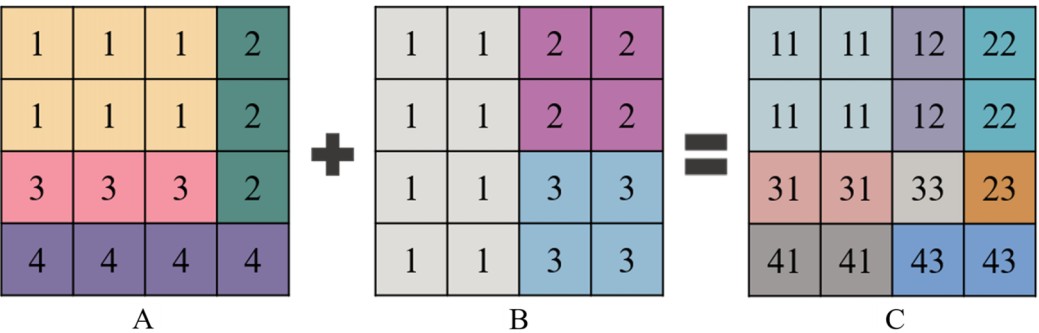

**Figure 4 Schematic diagram of merging over-segmentation regions of multi-temporal PolSAR images.** (A) Over-segmentation regions of single-temporal image A (four regions). (B) Over-segmentation regions of single-temporal image B (three regions). (C) Merging results of multi-temporal images (eight regions).

$$G_{h_s,h_r} = \frac{C}{h_s^2 h_r^3} g\left(\parallel \frac{x_s^2}{h_s} \parallel\right) g\left(\parallel \frac{x_r^2}{h_r} \parallel\right) \tag{2}$$

where $h_s$ and $h_r$ are window sizes for space vector and feature vector of each pixel, respectively, C is normalized constant, $g(\cdot)$ is a Gaussian function (*Zhao et al., 2015*). The detailed steps and parameter setting principles of PolSAR image segmentation using MS algorithm can be found in the literature *He et al. (2008)* and *Zou et al. (2009)*.

(2) Over-segmentation regions merging of multi-temporal images

Over-segmentation regions of multi-temporal images are merged by the over-segmentation regions of each single-temporal PolSAR image. The merging strategy is shown in Fig. 4. Each square describes a pixel in images, the value of the square means the attribute of the object in the images. Different values or colors in the squares discriminate the different objects in the images.

Even the over-segmentation result of the merged multi-temporal PolSAR image contains more regions than that of the single-temporal PolSAR image, the number of the regions in it is still less than the number of the pixels in the original image.

### Edge information extraction

The construction of similarity measurement matrix needs to select appropriate features as segmentation clues, which is not only the core of transforming image segmentation problem into graph segmentation problem, but also the premise of segmentation. Edge information is very important information in image segmentation, especially when excellent segmentation results can't be obtained by using information such as image gray level and color. For PolSAR images, the strong multiplicative noise makes the segmentation method based on pixel-based region growth unable to achieve application purpose, and relatively stable edge information is a excellent choice. Through the edge intensity information between pixels, the similarity between two pixels can be effectively measured. Therefore, the edge information of multi-temporal PolSAR images is used as the segmentation clue of SGP.

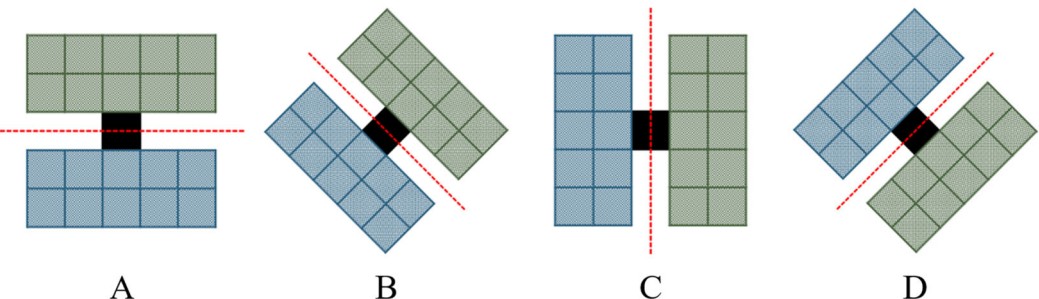

**Figure 5 Schematic diagram of edge detector.** Edge detectors are set in four directions. (A) 0° direction, (B) 45° direction, (C) 90° direction, (D) 135° direction.

(1) Edge information extraction of single-temporal

In this paper, referring to the method proposed by *Schou et al. (2003)* and *Zhao et al. (2015)*, edge detectors in four directions (0°, 45°, 90°, 135°) are set, as shown in Fig. 5. The window size can be determined according to the actual situation (3 × 3, 5 × 5, ...).

Assuming that the central pixel of the edge detector is the edge, there should be a strong difference between the regions on both sides of the central pixel. Therefore, the Wishart test statistics suitable for PolSAR data is utilized to measure the difference between two regions. In the Wishart algorithm, the maximum likelihood-ratio function is utilized to test the equality of center covariance matrices of two regions. The hypothesis test equation is

$$H_0 : \Sigma_i = \Sigma_j \quad versus \quad H_1 : \Sigma_i \neq \Sigma_j \tag{3}$$

where $\Sigma_i$ and $\Sigma_j$ are the center covariance matrices of the $i$th and $j$th regions, respectively. Let $\Theta_i$ and $\Theta_j$ be the sample covariance matrix data sets of the $i$th and $j$th regions, respectively. It is assumed that sample covariance matrices are spatially independent, therefore, the difference measure between the $i$th and $j$th regions can be derived from the likelihood ratio test. The test statistic equation is

$$Q = \frac{L_{H_0}\left(\hat{\Sigma}|\Theta_i, \Theta_j\right)}{L_{H_1}\left(\hat{\Sigma}_i, \hat{\Sigma}_j|\Theta_i, \Theta_j\right)} = \frac{\left|\hat{\Sigma}_i\right|^{nN_i}\left|\hat{\Sigma}_j\right|^{nN_j}}{\left|\hat{\Sigma}\right|^{n(N_i+N_j)}} \tag{4}$$

where $L_{H_0}(\cdot)$ and $L_{H_1}(\cdot)$ are likelihood functions under different assumptions, respectively. $\Sigma$ is the center covariance matrices of the whole region. $\hat{\Sigma}$, $\hat{\Sigma}_i$ and $\hat{\Sigma}_j$ are maximum likelihood estimators of $\Sigma$, $\Sigma_i$ and $\Sigma_j$, respectively. $N_i$ and $N_j$ are the numbers of samples in the $i$th and $j$th regions, respectively. $n$ is the number of looks. If the value of $Q$ is too low, the null hypothesis $H_0$ is rejected. Thus, the difference measure between the $i$th and $j$th regions can be defined as

$$D(S_i, S_j) = -\frac{1}{n}lnQ = (N_i + N_j)\ln\left|\hat{\Sigma}\right| - N_i ln\left|\hat{\Sigma}_i\right| - N_j ln\left|\hat{\Sigma}_j\right| \tag{5}$$

where $S_i$ and $S_j$ represent the $i$th and $j$th regions, respectively. The difference measure $D$ is symmetric. If $i = j$, $D(S_i, S_j)$ has a minimum value equaling to zero. If the $i$th and $j$th

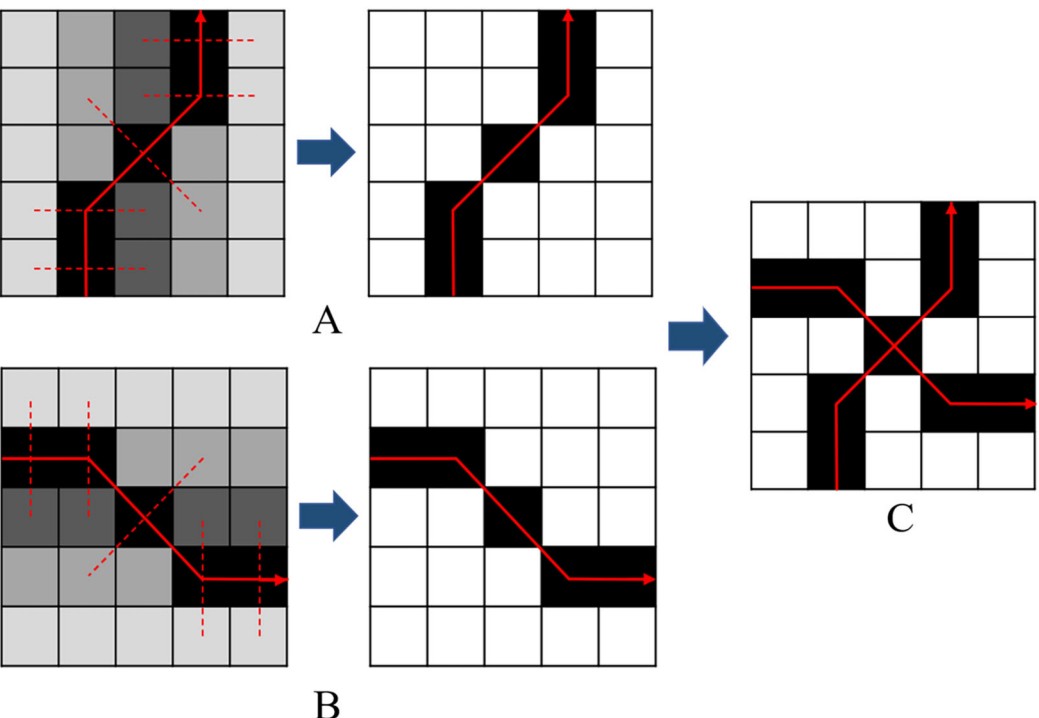

**Figure 6 Schematic diagram of single-temporal edge optimization and multi-temporal edge fusion.**
(A) Edge optimization schematic diagram of single-temporal A. (B) Edge optimization schematic diagram of single-temporal B. (C) Schematic diagram of multi-temporal edge fusion.

regions are more dissimilar, the value of $D(S_i, S_j)$ is higher. Details of the derivation of edge extraction algorithm can be found in, *Cao et al. (2007)* and *Liu et al. (2013)*.

The Eq. (5) is used to calculate the $D(S_i, S_j)$ in four directions of each pixel, that is, the edge intensity values in different directions, and retain the maximum edge intensity value $D_{max}$ and its edge direction $\theta^*$. After edge extraction, the polarimetric information of PolSAR image is transformed into edge information.

(2) Multi-temporal edge information fusion

First, it is necessary to further optimize the edge extraction results of the single-temporal PolSAR image to determine the boundary elements and avoid positioning deviations. For any pixel $x$ on the image, the maximum edge intensity value $D_{max}(x)$ and edge direction $\theta^*(x)$ orientation have been determined. Compare the edge intensity values of the pixels on both sides perpendicular to the edge direction $\theta^*(x)$. If the maximum edge intensity value $D_{max}(x)$ of pixel $x$ is greater than or equal to the edge intensity value of pixels on both sides, then keep the value, otherwise, set to zero. For example, as shown in Figs. 6A and 6B, assuming that $\theta^*(x)$ is 0°, $D_{max}(x)$ can be preserved only if it is greater than or equal to the edge intensity values of the upper and lower pixels. Similarly, if the $\theta^*(x)$ is 90°, the $D_{max}(x)$ can be retained only if it is greater than or equal to the edge intensity values of the left and right pixels. Then, based on the edge optimization results of different time temporal, the edge intensity values of pixels in the same position are

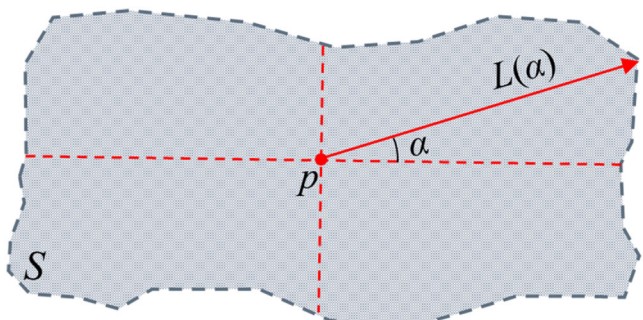

**Figure 7 The centre location of over-segmentation regional.** $S$ is a segmentation region, $p$ is the internal pixel of an over-segmentation region $S$, $\alpha$ is the azimuth angle, $L(\alpha)$ is the distance from the pixel p to the boundary of the region when the azimuth angle is $\alpha$.

compared, and the maximum value is taken as the multi-temporal edge value, that is, the edge information fusion of multi-temporal PolSAR images is completed. The fusion result is shown in Fig. 6C. The result of multi-temporal edge information fusion will provide segmentation clues for subsequent SGP of multi-temporal PolSAR images.

### Construction of similarity measurement matrix

On the basis of dividing the original pixels of the multi-temporal PolSAR image into many over-segmentation region units with similar statistical characteristics, it is necessary to find a pixel that can represent the spatial position of the entire over-segmentation region. The calculation method is shown in Fig. 7.

In the Fig. 7, $p$ is the internal pixel of an over-segmentation region $S$, $L(\alpha)$ is the distance from the pixel $p$ to the boundary of the region when the azimuth angle is $\alpha$, when the value of $\alpha$ is determined, the extensibility of the pixel $p$ in the over-segmentation region $S$ can be expressed as:

$$E(p \in S) = \prod_{i=1}^{n} L(\alpha \times i), \quad n = 2\pi/\alpha \tag{6}$$

where the pixel corresponding to $\max\{E(p \in S)\}$ is the pixel representing the $S$ space position of the entire over-segmentation region. If the over-segmentation region is a regular shape (such as a rectangle, a circle, etc.), the pixel corresponding to $\max\{E(p \in S)\}$ is the geometric centre of the region.

After obtaining the representative central pixel of the over-segmentation region of the multi-temporal PolSAR image, then, the edge fusion information of the multi-temporal PolSAR image can be used to measure the degree of similarity between the regions. The basic idea is shown in Fig. 8.

Figure 8A shows the Pauli RGB display of the PolSAR image, Fig. 8B is the local region, $S_1$, $S_2$ and $S_3$ are three segmentation regions. $p_1$, $p_2$ and $p_3$ are the corresponding spatial representative pixels. The similarity measure between the two regions can be represented by the maximum edge intensity value on the line between the representative

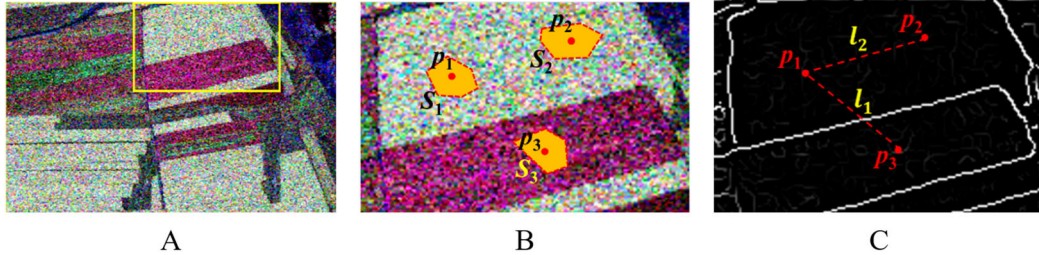

| A | B | C |

**Figure 8 Illustration of extracting the dissimilarity information from edge maps.** (A) A subset of a C-band PolSAR image (Pauli RGB). (B) Part of the original image marked by the box; $S_1$, $S_2$ and $S_3$ are three over-segmentation regions; $p_1$ and $p_2$ are in the same class, and $p_1$ and $p_3$ are in different classes. (C) Edge map of image (B) after the oriented nonmaximal suppression; somewhere along $l_1$, the value of edge map strength is high, which suggests that $p_1$ and $p_3$ are in different classes; along $l_2$, the values of edge map strength are all low, proposing that $p_1$ and $p_2$ are probably in the same class.

pixels, as shown in Fig. 8C. There are obvious edges between pixels $p_1$ and $p_3$, and the similarity will be lower. Correspondingly, the similarity between pixel $p_1$ and $p_2$ is higher.

Therefore, based on multi-temporal edge fusion information, the difference between any two pixels $x$, $y$ can be defined $D_c(x, y)$. As shown in the following equation (*Leung & Malik, 1998*; *Ersahin, Cumming & Ward, 2010*; *Ersahin, Cumming & Ward, 2014*).

$$D_c(x, y) = D^*(z^*), \quad z^* = \arg\max_{z \in l} D^*(z) \tag{7}$$

where $D^*(\cdot)$ denotes the strength of edge maps after the oriented nonmaximal suppression, $l$ is the line joining $x$ and $y$, and $z^*$ is the location where the strength of edge maps after the oriented nonmaximal suppression is maximum along $l$. Then, the pairwise affinity is defined using a gaussian kernel (*Shi & Malik, 2000*; *Ersahin, Cumming & Ward, 2010*; *Ersahin, Cumming & Ward, 2014*).

$$W(x, y) = \exp\left\{\frac{-D_C^2(x, y)}{2\sigma_C^2}\right\} \tag{8}$$

where $\sigma_C$ is the scaling parameter for the kernel. Based on this, the construction of the similarity measurement matrix between the over-segmentation regions of the multi-temporal PolSAR image can be completed.

### Normalized cuts

In the normalized cuts algorithm, *Shi & Malik (2000)* formulated visual grouping as a graph partitioning problem. The basic principle of graph-based partitioning schemes is to represent a set of points in an arbitrary feature space using an undirected graph $G = \{V, E\}$, where $V$ is for the vertices and $E$ is for the edges between the vertices. Each vertex corresponds to a point in the feature space, and the edge between two vertices, *e.g.*, $x$ and $y$, is associated with a weight $W(x, y)$, that indicates the affinity of the pair. Image segmentation can be formulated as the best partitioning of the feature space into two regions, $A$ and $B$, based on the minimum cut criterion. The cost function *cut* as follows is minimized
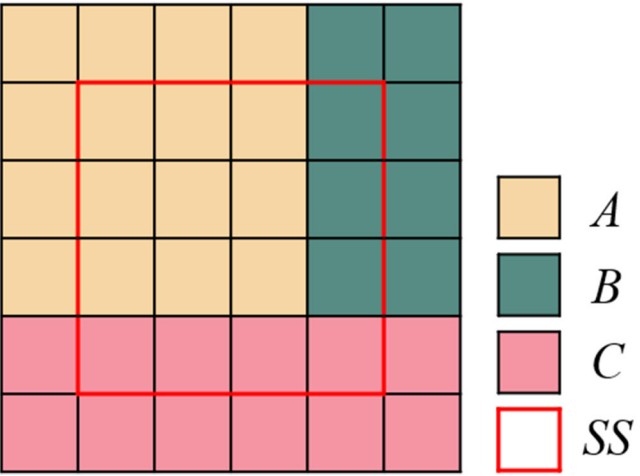

**Figure 9 Illustration chart of segmentation evaluation principle.** *SS* is the reference segmentation region; *A*, *B* and *C* are the segmentation regions, respectively.

$$cut(A, B) = \sum_{x \in A, y \in B} W(x, y) \tag{9}$$

where $W(x, y)$ is the pairwise affinity between $x$ and $y$.

Since minimizing the cost function in Eq. (9) favors cutting out small and isolated partitions, *Shi & Malik (2000)* proposed a new measure of partitioning cost. Instead of using value of total edge weight connecting the two regions, the new measure defines the partitioning cost as a fraction of the total edge connections to all the nodes in the graph. It is referred to as the normalized cut, *Ncut* as follows

$$Ncut(A, B) = \frac{cut(A, B)}{assoc(A, V)} + \frac{cut(B, A)}{assoc(B, V)} = \frac{cut(A, B)}{\sum_{x \in A, v \in V} W(x, v)} + \frac{cut(B, A)}{\sum_{y \in B, v \in V} W(y, v)} \tag{10}$$

where $A \cup B = V$, and $assoc(A, V)$ is the total connection from nodes in $A$ to all nodes in the graph and $assoc(B, V)$ is similarly defined. The principle of the normalized cuts algorithm is to minimize the partitioning cost measure *Ncut*. How to solve the minimization problem and more details related to the normalized cuts algorithm can be found in *Shi & Malik (2000)*.

### Method of segmentation evaluation

At present, there are many methods of segmentation evaluation, each of which has different advantages, disadvantages and applicable objects. In this paper, the improved method of maximum overlapping area proposed in reference *Zhao et al. (2015)* is used to evaluate the results of image segmentation. The segmentation accuracy calculated by this method reflects the degree of over-segmentation and under-segmentation of the image. The basic principle is shown in Fig. 9.

Firstly, for a reference segmentation region *SS*, there may be multiple segmentation regions to be evaluated ($A, B, C \ldots$) overlapping with it. Assuming that the overlapping region of the segmented region to be evaluated *A* and *SS* is the largest, the overlapping

region is recorded as $A_{SS}$, then $(1 - A_{SS}/A)$ is defined as the under segmentation ratio (USR). Then, the percentage of correctly segmented pixels can be calculated by limiting the value of USR. That is, only the pixels with the largest overlap region and the under-segmentation ratio not greater than USR will be counted as the number of pixels for correct segmentation. For example, as shown in Fig. 9, if we set USR = 0.3, then $(1 - A_{SS}/A) = (1-9/16) = 0.44 > 0.3$, it can be considered that the $A_{SS}$ region is a wrongly segmented. At this time, there is no correctly segmented pixel for the reference region SS, and its segmentation accuracy is 0. When setting USR = 0.5, $(1 - A_{SS}/A) = (1-9/16) < 0.5$, the $A_{SS}$ region can be considered as the correct segmented pixel. For the reference region SS, the segmentation accuracy is the ratio of the number of pixels in the region $A_{SS}$ to the number of pixels in the region SS. Finally, the segmentation accuracy of the whole image is the average of the segmentation accuracy of all reference segmentation regions.

## RESULTS

### Segmentation experiment

The PolSAR images of three temporal are pre-processed by radiometric calibration, image registration, multi-look, filter and geocode. First of all, the main purpose of radiometric calibration in PolSARpro6.0 software (https://earth.esa.int/web/polsarpro/home) is to determine the relationship between the grayscale of radar image and the standard backscatter coefficient. Secondly, the registration of multi-temporal images is carried out in Gamma software (https://www.gamma-rs.ch/). The detailed registration process can be found in the literature *Wang (2013)*. The image after registration is processed with $2 \times 2$ window multi-look processing in PolSARpro6.0 software, the pixel sizes in azimuth and range of PolSAR image after multi-look are about 9.92 and 9.47 m respectively. In order to reduce the influence of speckle noise, this paper uses Mean-Shift filtering method to filter, and its window is set to $5 \times 5$. Finally, the digital elevation model (DEM) data of 30 m resolution in the study area is used for geocoding in Gamma software. The projection of Radarsat-2 image is transformed into "UTM_Zone_51N" projection under WGS84 coordinate system, and resampled to obtain PolSAR polarization covariance matrix data with a spatial resolution of 10 m.

Then, the single-temporal PolSAR images are over-segmented by MS. In this paper, based on the existing literature reference (*He et al., 2008*; *Zou et al., 2009*; *Zhao et al., 2015*). The kernel function window values $h_s$ and $h_r$ of pixel space vector and feature vector are set to 7 and 6.5. And the minimum segmentation region parameter M plays a role in controlling the size of the segmentation region. The setting of this parameter only needs to ensure the over-segmentation degree of the image, which has a large setting space. In this paper, considering the operation cost of SGP for multi-temporal PolSAR images, a certain degree of attempt and comparison is made according to experience, and the segmentation results are qualitatively evaluated based on visual evaluation. The parameter M which ensures that each of the three temporal PolSAR images can be over-segmented and has a small computational cost is determined as the optimal parameter selected in this paper. As shown in Figs. 10A–10C, the number of over-segmentation regions of MS

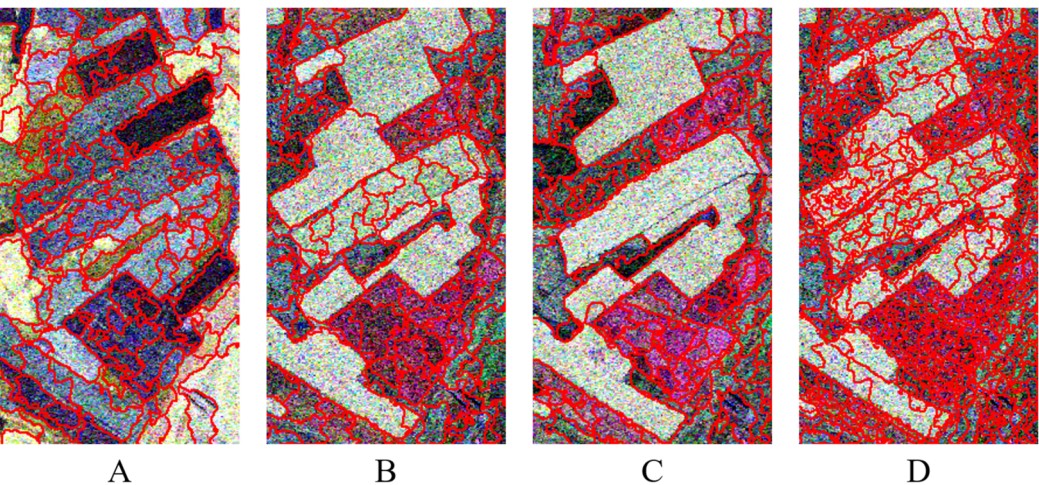

**Figure 10 MS pre-segmentation results of single-temporal PolSAR images and merging results of multi-temporal segmentation regions (A–D).** The $h_s$ and $h_r$ of the three temporal are set to 7 and 6.5, respectively; M is set to 235, 235 and 192, respectively; and the number of blocks in the over-segmentation region of MS is 118, 102 and 111, respectively. (D) The merging results of multi-temporal segmentation regions is 1,411, and the Pauli RGB of 20130803 is used as the display image. (A) 20130523, $h_s = 7$, $h_r = 6.5$, M = 235, N = 118; (B) 20130803, $h_s = 7$, $h_r = 6.5$, M = 235, N = 102; (C) 20130827, $h_s = 7$, $h_r = 6.5$, M = 192, N = 111; (D) multi-temporal, N = 1,411.

for different temporal is 118, 102 and 111, respectively, under setting of different M (M: 235, 235, 192). Based on this, the over-segmentation results produced by the initial segmentation of three temporal PolSAR images are merged, and the number of over-segmentation regions after merging is 1411. The merging result of multi-temporal over-segmentation regions is shown in Fig. 10D, with the 20130803 Pauli RGB as the display image.

In addition, the central position of the over-segmentation region is calculated according to Eq. (6). That is, the pixels that can represent the spatial position of the over-segmentation region of the single-temporal and multi-temporal PolSAR images are obtained. The result is shown as the yellow point in Fig. 11. It can be seen that these points are in the relative center of the region.

While the over-segmentation of multi-temporal PolSAR images is carried out, the edge information of multi-temporal PolSAR images is also extracted to obtain the segmentation clues of SGP. Firstly, the edge information of the single-temporal PolSAR image is extracted based on the edge detector, and the window is set to 7 × 7. Then, edge optimization based on rough edge extraction. Based on the edge optimization results of different temporal, the edge intensity values of pixels in the same position are compared, and the maximum value is taken as the multi-temporal edge value. As shown in Fig. 12. Among them, the edge information of the farmland parcels in the yellow rectangle in Fig. 12A, the parcels in the green rectangle in Fig. 12B and the parcels in the red rectangle in Fig. 12C are clearer and more complete compared with the same position of the other two images. The edge information of the three temporal PolSAR images is fused, so that the result after the fusion Fig. 12D contains rich edge information of the multi-

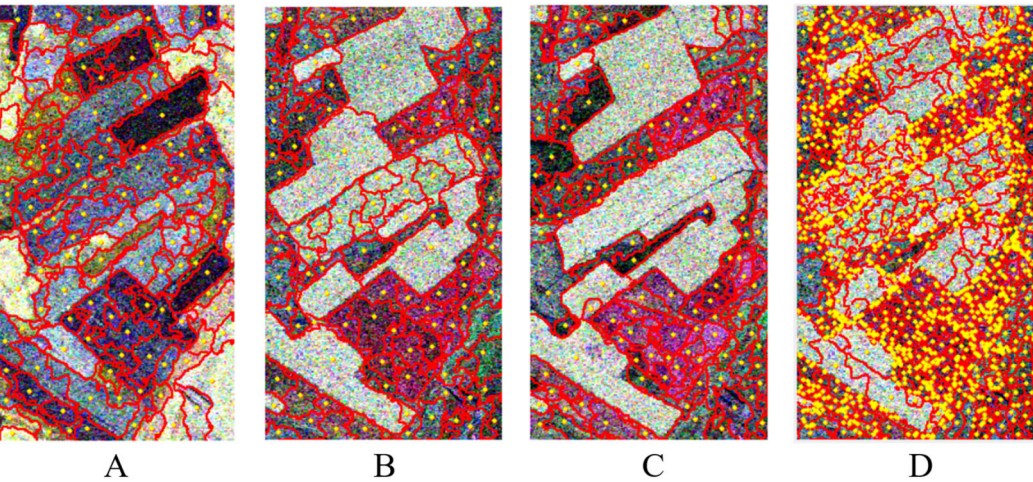

**Figure 11 Over-segmentation regional center of single-temporal and multi-temporal PolSAR images.** (A) 20130523; (B) 20130803; (C) 20130827; (D) multi-temporal.

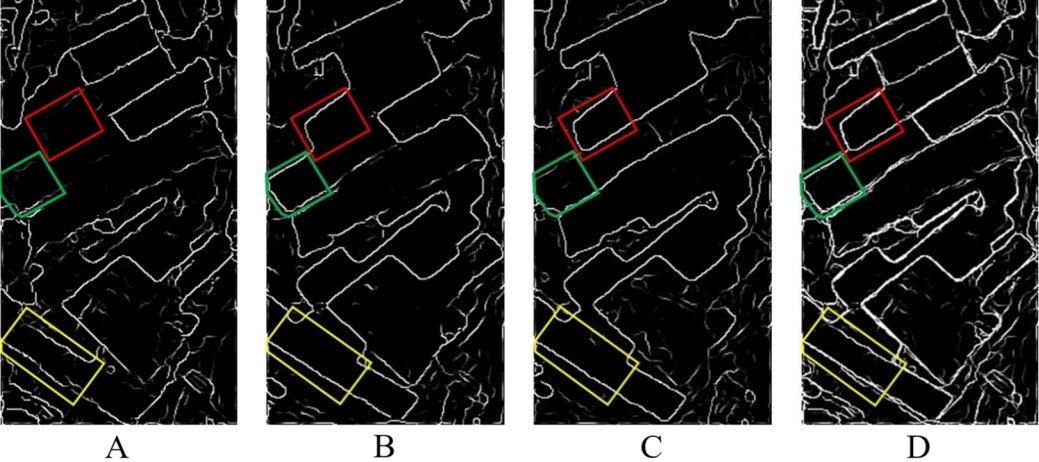

**Figure 12 Single-temporal PolSAR image edge extraction results and multi-temporal edge fusion results.** (A) 20130523; (B) 20130803; (C) 20130827; (D) multi-temporal edge fusion results.

temporal PolSAR image. The edge fusion results are used as the segmentation clues for the subsequent SGP, which can achieve better segmentation results than single-temporal PolSAR images.

Through the above-mentioned pixel points representing the spatial position of the over-segmentation regions of the multi-temporal PolSAR image Fig. 11D and the edge fusion result of the multi-temporal PolSAR image Fig. 12D, the similarity measurement matrix can be constructed. Then, the segmentation of the multi-temporal PolSAR image is completed by the normalized cut criterion. In this step, the segmentation scale can be controlled by setting the number of final segmentation regions. After a certain degree of attempt and comparison according to the experience, and the qualitative evaluation of the segmentation results based on visual evaluation, it can be found that the optimal

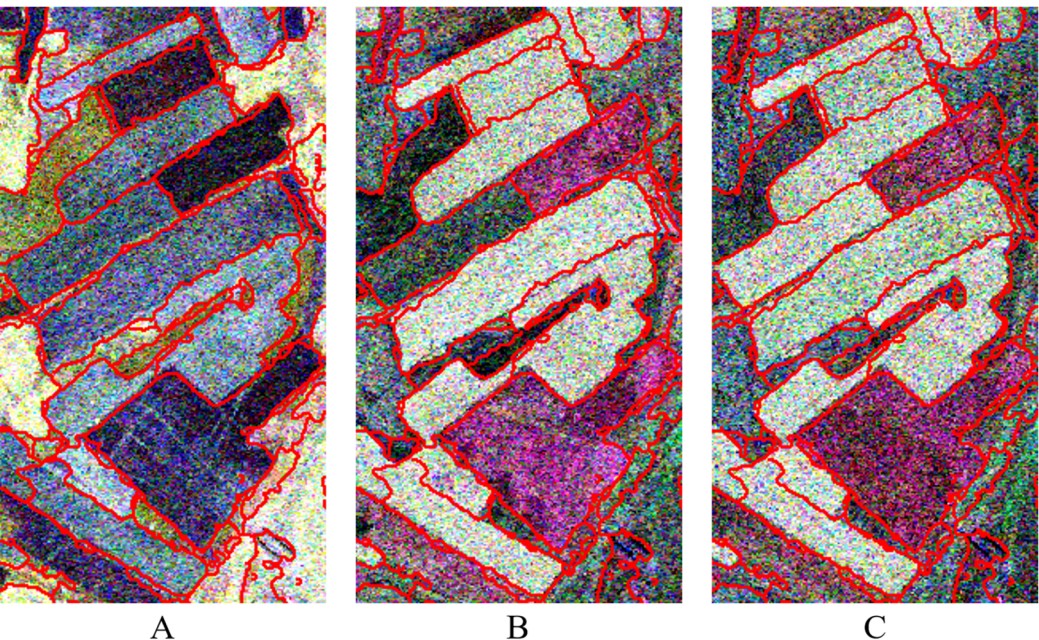

A          B          C

**Figure 13 Segmentation results of multi-temporal PolSAR images.** The number of segmentation regions is set to 47. (A) 20130523, $N = 47$; (B) 20130803, $N = 47$; (C) 20130827, $N = 47$.

segmentation effect can be achieved when the number of segmentation regions is set to 47. At this time, the farmland parcels of different sizes are well segmented and become independent objects. In addition, the details of segmentation results are excellent, and the regional consistency is well maintained. The final segmentation result is shown in Fig. 13. In order to show the effect of the multi-temporal MS-SGP segmentation algorithm, the PauliRGB of three temporal PolSAR images are all used as the display image, respectively.

In terms of the operation cost of the algorithm, the experimental image consists of 374 rows and 205 columns, with a total of 76,670 pixels. If the traditional pixel-based SGP is adopted to complete the segmentation of multi-temporal PolSAR images, the similarity measurement matrix of 76,670 × 76,670 size needs to be established through $C^2_{76670}$ operations, while the proposed method only needs $C^2_{1411}$ operations to establish a similarity measurement matrix of 1,411 × 1,411 size. Thus, the requirements of computing space and time are reduced by about 3,000 times. It can be seen that the pre-segmentation with MS can reduce the operation cost of SGP to a certain extent, and the global optimization strategy of SGP also is used to achieve excellent segmentation effect.

## Quantitative evaluation and analysis

In the previous section, multi-temporal MS-SGP segmentation algorithm is used to obtain the segmentation results of multi-temporal PolSAR images. In order to evaluate the segmentation algorithm more objectively, a quantitative method is used to evaluate the pros and cons of the segmentation results. Firstly, the accuracy of the segmentation results of multi-temporal PolSAR images are evaluated based on the obtained segmentation
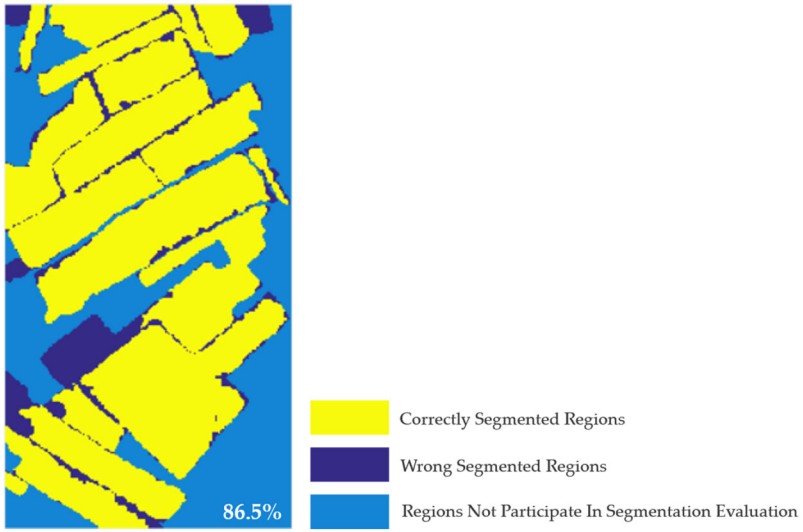

86.5%

Correctly Segmented Regions

Wrong Segmented Regions

Regions Not Participate In Segmentation Evaluation

**Figure 14 Correct segmentation pixel distribution map of multi-temporal PolSAR image. (USR = 0.3).** The segmentation accuracy is 86.5%.

reference images (Fig. 2), as shown in Fig. 14. It is calculated that when USR = 0.3, the segmentation accuracy of multi-temporal PolSAR image is 86.5%. Fig. 14 is a distribution diagram of correctly segmentation pixels in multi-temporal PolSAR images. Among them, the yellow region is the pixel of correct segmentation, the indigo region is the pixel of wrong segmentation, and the blue region is the pixel that do not participate in segmentation evaluation. It can be seen that the multi-temporal MS-SGP segmentation algorithm proposed in this paper can achieve preferable segmentation results. The farmland parcel unit of the experimental region can be segmented relatively completely, making it an independent farmland parcel object. The regional consistency is excellent, which lays a good foundation for the next image interpretation.

In order to clarify the influence of the parameter setting of each link on the segmentation accuracy of the multi-temporal MS-SGP segmentation algorithm proposed in this paper, the variation of segmentation accuracy with the number of final segmentation regions and the number of MS over-segmentation regions is analyzed, as shown in Figs. 15 and 16. Finally, in order to further discuss the influence of the choice of temporal on the segmentation accuracy, the segmentation accuracy of different combinations of single-temporal and multi-temporal is compared, as shown in Fig. 17. Based on the above analysis, users can fully understand the performance of the algorithm.

The trend of the segmentation accuracy of the multi-temporal MS-SGP segmentation algorithm with the number of final segmentation regions is shown in Fig. 15. Except for the different settings of the final number of segmentation regions, the other parameter settings are consistent with those in Fig. 10. It can be seen that when the number of regions is in the range of 20–40, the segmentation accuracy shows an increasing trend with the increase of the number of regions. Then it reached a peak and stabilized in the range of 40–120 regions, and then showed a downward trend. It can be seen that the segmentation effect is the best and stable when the number of regions in the map is in the

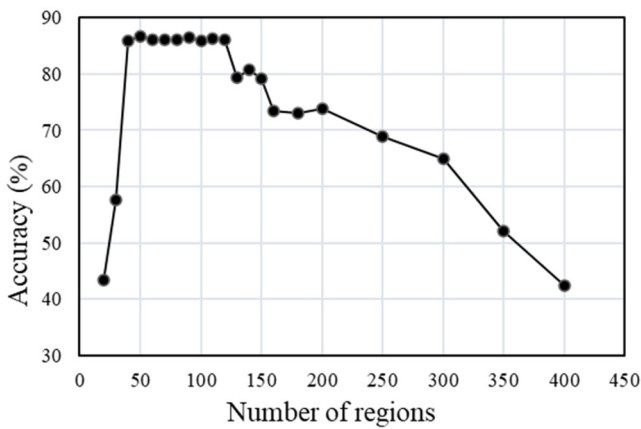

**Figure 15 Segmentation accuracy of multi-temporal MS-SGP segmentation algorithm changes with the number of regions.**

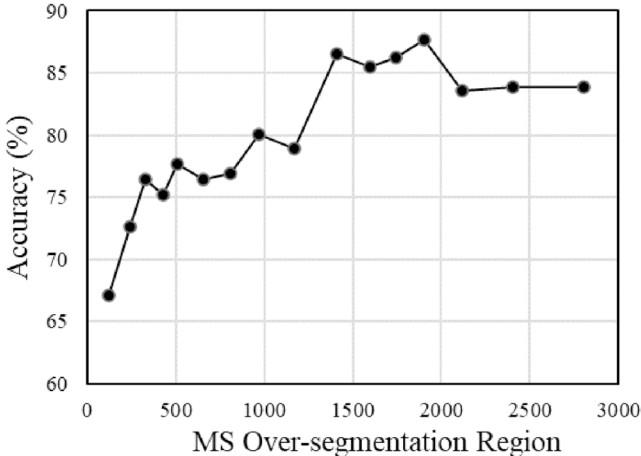

**Figure 16 Segmentation accuracy of multi-temporal MS-SGP segmentation algorithm changes with the merging number of MS over-segmentation regions.**

range of 40–120, which is the segmentation scale that users can choose. In addition, Fig. 15 also verifies that the segmentation evaluation index used in this article can reflect both over-segmentation and under-segmentation. Whether the number of regions is too small or too many, the segmentation accuracy will be low.

The trend of the segmentation accuracy of the multi-temporal MS-SGP segmentation algorithm with the number of merged MS over-segmentation regions is shown in Fig. 16. Except for the different number of merged MS over-segmentation regions, the other parameter settings are the same as those in Fig. 13. It can be seen that the segmentation accuracy exhibits a fluctuating growth trend with the increase of the number of regions and stabilizes after reaching a peak. It shows that the number of regions in the over-segmentation stage of MS should not be too small, and it is necessary to avoid the impact of under-segmentation on the subsequent segmentation effect. The specific parameter setting should firstly ensure the realization of over-segmentation, and secondly consider the influence of the number of regions on the operation cost.

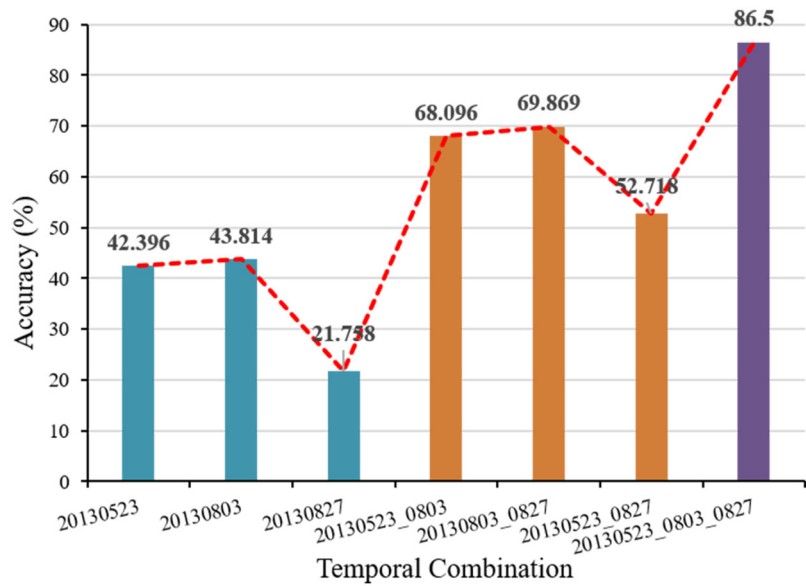

**Figure 17 Segmentation accuracy with different temporal combination.**

The variation trend of segmentation accuracy in the combination of temporal is shown in Fig. 17. The temporal data are three images acquired on May 23, August 3, and August 27, 2013. Except for the different temporal combination, the other parameter settings are the same as those in Figs. 10 and 13. It can be seen that the segmentation accuracy of the two-date combination is greater than that of the single-temporal, but is less than the segmentation accuracy of the three temporal combination. This shows that the addition of multi-temporal information can effectively improve the segmentation accuracy and obtain the ideal segmentation effect.

## Comparison and analysis of single-temporal segmentation results

In order to further verify the effectiveness of the proposed method, the single-temporal MS-SGP segmentation algorithm is used to segment three temporal PolSAR images. Furthermore, the segmentation results of multi-temporal MS-SGP segmentation algorithm and single-temporal MS-SGP segmentation algorithm are compared and analyzed. For the single-temporal PolSAR image, the representative pixels of spatial position of over-segmentation region (Figs. 11A–11C) and edge information (Figs. 12A–12C) of three temporal were used as input of SGP. The number of segmentation regions in the final SGP is consistent with that in Fig. 13, which is 47. The results are shown in Fig. 18.

Figure 18A shows the segmentation results of the first temporal (20130523) PolSAR image. Due to the most crops had just been sown when the images were obtained in this period, the backscattering in this region was mainly soil surface scattering. Therefore, it is easy to be segmented when the difference of soil backscattering coefficient between different farmland parcels is large. As shown by the farmland parcel in the black rectangle, it has different color representations from the surrounding field parcels, and the difference is obvious. At this time, the farmland parcel has been well segmented and has become an

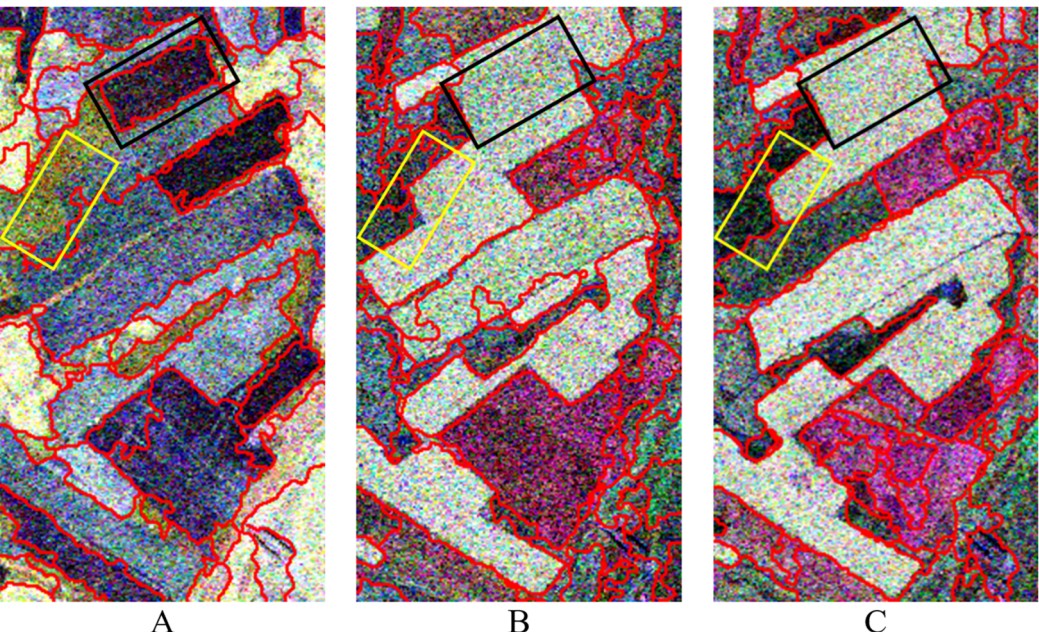

**Figure 18 Segmentation results of single- temporal PolSAR images.** The number of segmentation regions is set to 47. (A) 20130523, $N = 47$; (B) 20130803, $N = 47$; (C) 20130827, $N = 47$.

independent object. However, the farmland parcels at the same location in the two temporal images of Figs. 18B and 18C are not segmented. The reason is that the rape in the two images is in the filling stage and mature stage respectively, growing well and the vegetation structure is similar. A large region of rape planting farmland parcels have similar backscattering coefficients and have the same color representation on images. However, the ridge that actually distinguishes the boundary of the farmland parcels can't be reflected in the image due to its limited width (about 0.5 m) (*Zou, Li & Tian, 2014*). Therefore, it is not possible to effectively segment different field parcel units based on single-temporal PolSAR images.

Although the region in the yellow rectangle in Fig. 18A contains two types of ground targets, rape and shrubs. However, due to the small difference of backscattering coefficient and similar color representation of soil surface in this area, its separability is poor and it can't be effectively segmented. On the other hand, the rape and shrub in the same region of the two images of Figs. 18B and 18C were well segmented, because that rape grows vigorously in these two images, and its backscattering is stronger, which is obviously higher than other ground targets. At this time, the backscattering coefficients of ground targets and their color representations in images are quite different between rape and shrub, so rape and shrub can be well separated.

Based on the obtained segmentation reference images (Fig. 2), the accuracy of segmentation results of single-temporal PolSAR images is evaluated, as shown in Fig. 19. It is calculated that when USR = 0.3, the segmentation accuracy of single-temporal PolSAR image are 42.4%, 43.8% and 21.8%, respectively. It can be seen that the segmentation results of single-temporal PolSAR images show a large area of wrong

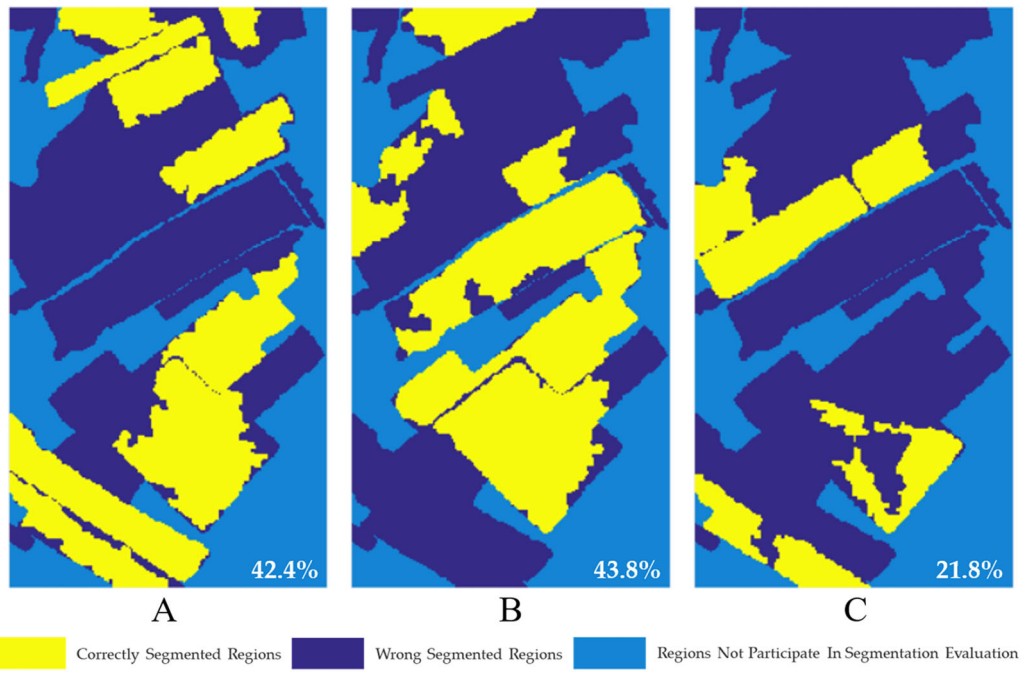

**Figure 19 Correct segmentation pixel distribution map of single-temporal PolSAR image.
(USR = 0.3).** The segmentation accuracy are 42.4%, 43.8% and 21.8%, respectively.

segmentation, and the final segmentation accuracy is poor. It can be seen that the segmentation effect of the multi-temporal MS-SGP segmentation algorithm proposed in this paper (Fig. 14) is better than that of the single-temporal MS-SGP segmentation algorithm.

It is not difficult to see from Figs. 18 and 19. In each period of images, the difference of surface scattering, secondary scattering and volume scattering between two types of objects or multiple types of objects is small, so it is difficult to effectively identify all objects by using single-temporal PolSAR images (*Deng et al., 2014*). Through the segmentation of multi-temporal PolSAR images, it is found that the backscattering coefficients of some two types of objects may be close in one period of images, but quite different in the other period of images. Therefore, it is beneficial to improve the segmentation accuracy to try to combine multi-stage images and make comprehensive use of the characteristics of images in different periods for multi-temporal MS-SGP segmentation. At the same time, it can distinguish the planting time sequence information and the growth status information of the ground targets which cannot be identified in the traditional algorithm, which is of great significance for the monitoring of crops.

## DISCUSSION

Under the technical framework of SGP, this paper proposes a new algorithm for multi-temporal PolSAR image segmentation, which is used to solve the segmentation problems in some specific difficult scenes which can't be realized by single-temporal PolSAR image segmentation algorithm. Furthermore, through the same type of

comparative experiments between the previously proposed single-temporal MS-SGP segmentation algorithm by *Zhao et al. (2015)* and the multi-temporal MS-SGP segmentation algorithm proposed in this paper, the effectiveness of the proposed algorithm is verified. The experimental results show that the optimal segmentation is realized based on multi-temporal PolSAR images and combined with the global optimization of SGP algorithm. Thereby solving the problem that single-temporal PolSAR images and the super-pixels generated by local optimization algorithms both can't meet the needs of agricultural production. This lays a good foundation for the further interpretation of multi-temporal PolSAR image.

The algorithm constructs an edge detector based on the Wishart statistical distribution of polarization covariance data and hypothesis testing, and extracts the edge information of PolSAR image as the segmentation clue of SGP. Compared with the method proposed by *Ersahin, Cumming & Ward (2010)* to extract edge information by selecting amplitude or intensity images with different polarization channels and using edge detection operator, the proposed edge extraction algorithm can make full use of the full polarization information of PolSAR images. Among them, the strength of edge information depends on the difference of polarization information between targets. For single-temporal PolSAR images, there are cases in which different targets show similar polarization characteristics, and mis-segmentation is easy to occur when the edge information is weak. At this time, based on multi-temporal PolSAR images, the scattering characteristics of ground targets in different growth periods can be obtained, which can solve the above problems. In other words, this paper fuses the edge information of single-temporal PolSAR images to make them get useful supplement to each other, so as to obtain more complete and clear edge fusion results than any single-temporal PolSAR images, which can be used as the segmentation clue of subsequent multi-temporal PolSAR images SGP. The targets which can't be distinguished by single-temporal PolSAR image segmentation algorithm can be segmented. However, this method also has some shortcomings. The registration accuracy of multi-temporal PolSAR images directly affects the result of edge fusion. In addition, this process will inevitably introduce some noise problems, so that the multi-temporal edge fusion results have a certain degree of "pepper and salt" phenomenon, which is also reflected in the final segmentation results (Fig. 13). Therefore, in the next research, texture and spatial features should be comprehensively considered, so as to explore whether the introduction of other features is helpful to the segmentation of multi-temporal PolSAR images.

In this paper, the experiment of multi-temporal MS-SGP segmentation of crops with time series changing characteristics was carried out by using three temporal Radarsat-2 images. Compared with the local optimization segmentation algorithms such as level set (*Zou, 2015*), simple nonlinear iterative clustering (*Ma et al., 2021*) and binary partition trees (*Alonso-Gonzalez, Lopez-Martinez & Salembier, 2014*) currently applicable to multi temporal PolSAR images, the proposed algorithm makes use of the advantages of SGP algorithm, which can cluster in arbitrary shape sample space and converge to the global optimal solution, the optimal segmentation of farmland parcel object to meet the needs of agricultural production is realized. Although in terms of classification, change detection

and other applications, the super-pixels generated by the local optimization algorithm have met the application requirements. However, for agricultural applications, as the smallest unit of farmers' production and management, accurate identification of farmland parcels is conducive to the realization of crop production planning, management and benefit evaluation. However, the shape feature of the over-segmented object based on homogeneous pixel clustering is not meaningful, nor does it have any social attributes, that is, it is not convenient for production and application; In addition, the polarization features extracted from farmland parcel units can avoid the influence of outliers caused by mixed pixels and internal variation, and are more accurate than those extracted by over-segmented objects. Thus, it is more beneficial to carry out applications such as crop target identification. Many studies have also shown that the parcel-oriented crop classification method has higher classification accuracy than the over-segmented object-oriented classification method. Therefore, according to the characteristics of crop planting structure, many scholars adopt the classification method of taking farmland parcel as the basic unit to improve the classification accuracy. However, how to determine the optimal segmentation result still needs to select its relevant parameters according to experience. Although the setting range of related parameters is relatively wide, it still needs to go through a certain degree of attempt and comparison to achieve the optimal segmentation effect. Therefore, how to choose the optimal parameter configuration according to the image characteristics or develop a segmentation algorithm with adaptive parameters is the focus of in-depth research.

However, due to the difficulty of obtaining real verification data of multi-temporal PolSAR images, this paper is only discussed in a relatively small study area. In the follow-up, we will try to extend the algorithm to the practical application of large areas, which is of great significance and value to solve the problems of farmland parcel boundary extraction and target recognition in large crop planting areas. However, the application of this algorithm to a large research area will bring high computational cost to a certain extent, so it is necessary to use a high-performance computer or block the image. Moreover, more prior knowledge will be needed to determine the correlative parameter. In addition, there will be higher requirements for the registration accuracy of multi-temporal PolSAR images, so as to avoid excessive influence of "salt and pepper" phenomenon on the results of multi-temporal edge fusion. Therefore, the application of the algorithm to large research areas is the focus of in-depth research in the future.

## CONCLUSIONS

Aiming at the problem that the single-temporal PolSAR image segmentation algorithm is difficult to provide correct segmentation results for applications such as target recognition and time series analysis of ground targets with time series changes. This paper proposes a new algorithm for multi-temporal PolSAR image segmentation, and uses the three temporal Radarsat-2 PolSAR data to verify the effectiveness of this algorithm. The experimental results show that the multi-temporal MS-SGP segmentation algorithm proposed in this paper comprehensively utilizes the abundant ground targets polarimetric and temporal information of multi-temporal PolSAR images. The algorithm can segment

the ground targets that can't be distinguished using single-temporal PolSAR images and realize the accurate identification of farmland parcel units. At the same time, the segmentation accuracy of this method reaches 86.5%, which is significantly improved compared with the single-temporal PolSAR image segmentation accuracy, and the segmentation result is closer to the segmentation result of reference image.

The algorithm proposed in this paper still has some shortcomings, such as the selection of optimal parameters in the whole segmentation algorithm, the applicability of the algorithm, and whether other features can be introduced, which need to be further studied.

## ACKNOWLEDGEMENTS

We are also grateful for the technology and SAR data support provided by the Institute of Forest Resource Information Techniques, Chinese Academy of Forestry.

### Funding

This research was funded by the National Natural Science Foundation of China (41801289), the National Science and Technology Major Project of China's High Resolution Earth Observation System (21-Y20B01-9001-19/22) and the National Natural Science Foundation of China (42161059; 31860240; 32160365). The funders had no role in study design, data collection and analysis, decision to publish, or preparation of the manuscript.

### Grant Disclosures

The following grant information was disclosed by the authors:
National Natural Science Foundation of China: 41801289.
National Science and Technology Major Project of China's High Resolution Earth Observation System: 21-Y20B01-9001-19/22.
National Natural Science Foundation of China: 42161059, 31860240, 32160365.

### Competing Interests

The authors declare that they have no competing interests.

### Author Contributions

- Caiqiong Wang performed the experiments, analyzed the data, authored or reviewed drafts of the paper, and approved the final draft.
- Lei Zhao conceived and designed the experiments, prepared figures and/or tables, and approved the final draft.
- Wangfei Zhang performed the experiments, analyzed the data, prepared figures and/or tables, authored or reviewed drafts of the paper, help analyze and polish the text, and approved the final draft.
- Xiyun Mu analyzed the data, prepared figures and/or tables, and approved the final draft.
- Shitao Li analyzed the data, prepared figures and/or tables, and approved the final draft.

## Data Availability

The original data and codes are available in the Supplemental Files.

## Supplemental Information

Supplemental information for this article can be found online at http://dx.doi.org/10.7717/peerj.12805#supplemental-information.

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
