# Peer review of "Segmentation of multi-temporal polarimetric SAR data based on mean-shift and spectral graph partitioning"

_PeerJ, doi:10.7717/peerj.12805_

## Round 0.1 · original submission · Major Revisions

The reviewers suggested a major revision. I concur with the comments. The authors must provide more details regarding the design and applications of the algorithm in image segmentation of difficult scenarios.

Reviewer 1 ·

Basic reporting

Please see the attached file.

Experimental design

Please see the attached file.

Validity of the findings

Please see the attached file.

Additional comments

Please see the attached file.

Annotated reviews are not available for download in order to protect the identity of reviewers who chose to remain anonymous.

·

Basic reporting

no comment

Experimental design

no comment

Validity of the findings

no comment

Additional comments

The article is good and well explained. A new algorithm for multi-temporal PolSAR image segmentation is proposed in this paper. It could be useful to solve the segmentation problem in some specific difficult scenes, e.g. cropland, where crops of different types and different sowing times are easy to show similar characteristics on single-temporal PolSAR images.

1. Line 297: More details in preprocessing steps should be given, such as window sizes, geographical coordinate system, software.
2. Line: 304: How to set the minimum segmentation region parameter M? The value of M (192) in the third date is different from values (235) in another two dates. What is the reason?
3. Line 339: The number of segmentation regions is set to 47. However, it can be seen from Figure 15 that the segmentation accuracy reached a peak at 50 and stabilized in the range of 40-120. Why don’t you choose 50 as the number of segmentation?
4. Line 386-388: The specific parameter setting should firstly ensure the realization of over-segmentation, and secondly consider the influence of the number of regions on the operation cost. How to set the specific parameter in practice. Could you give more details?
5. It is better to show the correct segmentation map with bright colors in Figures 14 and 19. In addition, the segmentation accuracy should be given in Figure 14.
6. The current discussion part presents segmentation accuracies with single-temporal PolSAR images and different temporal combinations. It is more like the result part. In the discussion part, the authors could discuss what are their specific research findings or some influence factors. Also, the relation of this study to previous studies to show its significance is high will be more convincing and it is recommended.

---

## Round 0.2 · accepted · Accept

The reviewers have found that your paper has been significantly improved and is suitable for publication. The algorithm you proposed in the paper using multitemporal data is interesting to the remote sensing science community.

Reviewer 1 ·

Basic reporting

The reviewer does not have further comments.

Experimental design

The reviewer does not have further comments.

Validity of the findings

The reviewer does not have further comments.

Additional comments

The reviewer does not have further comments.

·

Basic reporting

no comment

Experimental design

no comment

Validity of the findings

no comment

Additional comments

The authors have made proper revisions. I have no other comments.